# Characterization of Neonatal Infections by Gram-Negative Bacilli and Associated Risk Factors, Havana, Cuba

**Arlenis Oliva [1], Yenisel Carmona [2], Elizabeth de La C. López [3], Roberto Álvarez [3], Meiji Soe Aung [4], Nobumichi Kobayashi [4] and Dianelys Quiñones [2,*]**

1    William Soler Hospital, Havana 10800, Cuba; arlenisof@infomed.sld.cu
2    Tropical Medicine Institute "Pedro Kouri", Havana 11400, Cuba; yeniselc@ipk.sld.cu
3    Public Health Ministry, Havana 10400, Cuba; elizabeth.lopez@infomed.sld.cu (E.d.L.C.L.);
     rafumero@infomed.sld.cu (R.Á.)
4    Department of Hygiene, Sapporo Medical University School of Medicine, Sapporo 060-8556, Japan;
     meijisoeaung@sapmed.ac.jp (M.S.A.); nkobayas@sapmed.ac.jp (N.K.)
*    Correspondence: diany.quinones@infomed.sld.cu

**Abstract:** Infections represent an important problem in neonates because of the high mortality. An increase in neonatal infections has been found in Cuban hospitals in recent years. The aim of this study was to provide evidence on the clinical and microbiological behavior of Gram-negative bacilli that cause neonatal infections in hospitals of Havana, Cuba. It was carried out as a descriptive cross-sectional investigation from September 2017 to July 2018 in The Tropical Medicine Institute "Pedro Kouri" (IPK). Sixty-one Gram-negative bacilli isolated from neonates with infections in six Gyneco-Obstetric and Pediatric Hospitals of Havana were analyzed for their species and antimicrobial susceptibility. Late-onset infections were more common than early-onset ones and included urinary tract infection in the community (87%) and sepsis in hospitals (63.3%). Catheter use (47%) and prolonged stay (38%) were the most frequent risk factors. Species of major pathogens were *Escherichia coli* (47%) and *Klebsiella* spp. (26%). The isolated Gram-negative bacilli showed high resistance rates to third-generation cephalosporins, ciprofloxacin and gentamicin, while being more susceptible to carbapenems, fosfomycin, colistin and amikacin. The present study revealed the clinical impact of Gram-negative bacilli in neonatology units in hospitals of Havana. Evaluation of antimicrobial susceptibilities to the isolates from neonates is necessary for selection of appropriate empirical therapy and promotion of the rational antibiotic use.

**Keywords:** neonatal infection; gram-negative bacilli; antimicrobial resistance; Cuba

## 1. Introduction

Neonatal infection is the pathological process that is caused by the invasion of pathogenic or potentially pathogenic microorganisms in normally sterile tissues, fluids or body cavities. According to the time of onset, it is classified into early or late neonatal infection [1].

The most serious manifestation is sepsis, with a lethality over 50%. The incidence of neonatal sepsis in the developed world is between 0.6% and 1.2% of all live born babies, but, in the developing world, it can reach from 20% and 40% of all live born babies [2].

The postnatal period is risky for the multiple opportunities of exposure to pathogenic microorganisms. Prematurity, low birth weight, exposure to invasive procedures, receiving parenteral nutrition with lipid emulsions, alterations in the skin and/or mucous membrane barriers, frequent use of broad-spectrum antibiotics and prolonged hospital stay are the most common risk factors for infections in newborns [3].

According to estimates by the World Health Organization (WHO) in 2017, 46% of deaths in children under fiveyears occurred in newborns, with infections being ranked as

the second direct cause of death [3]. The rates of neonatal infection vary depending on geographic region, economic resources, access to health andmaternal and fetal risks. In Latin America, the incidence of sepsis ranges between 3.5% and 8.9%, while, in industrialized countries such as the USA, this incidence is reported between 1 and 5 cases per 1000 live newborns [4].

Neonatal infection of bacterial etiology is mainly caused by Gram-positive cocci and Gram-negative bacilli. In the latter group, dominant species are members of *Enterobacteriaceae*, such as *Escherichia coli*, *Klebsiella* spp., *Enterobacter* spp. and *Serratia* spp. as well as non-fermenting bacilli (BNF) such as *Pseudomonas aeruginosa* and *Acinetobacter baumannii* [4].The treatment of Gram-negative bacilli infections represents the biggest challenge for neonatologistsbecause of their ability to develop antimicrobials resistance, which may bring difficulty in choice of treatment [5].

In Cuba, the early neonatal mortality rate was 1.6 and the late neonatal 0.7 per 1000 live born babies in 2016, and infections represent the third cause of mortality [6]. In specific studies on neonatal sepsis in the country, *Klebsiella pneumoniae*, *Escherichia coli*, *Enterobacter* spp. and *Serratia* spp. were the main etiologic agents of hospital infections with Gram-negative bacilli [7].

Research providing evidence of the behavior of neonatal infection by Gram-negative bacilli and their susceptibility patterns is scarce in Cuba. Therefore, the National Reference Laboratory of Healthcare Associated Infections of the Tropical Medicine Institute "Pedro Kourí" (NRL-HCAI/IPK) conducted research on neonatal infections, to obtain evidence on the clinical and microbiological features of Gram-negative pathogens causing neonatal infections in hospitals in Havana, which may determine more frequent clinical forms of the infection and describe risk factors associated with neonatal infection and species of Gram-negative bacilli and their antimicrobial susceptibility.

## 2. Materials and Methods

A descriptive cross-sectional study was carried out in NRL-HCAI/IPK from September 2017 to July 2018. Sixty-one Gram-negative bacilli isolated from neonates with infection in six Gyneco-Obstetric and Pediatric Hospitals of Havana were analyzed. The following hospitals participated to this study: Juan Manuel Márquez, Ramón González Coro, William SolerLedea, Daughters of Galicia, Eusebio Hernández and Enrique Cabrera. Only one isolate per patient was included in this study.

For the collection of clinical and epidemiological information, all records of infants infected with Gram-negative bacilli were confirmed during this study period; all of them were supervised by the Head of the Neonatology Services. Records were requested from the corresponding hospital files with prior authorization from the Ethics Committee.

In the present study, healthcare-associated infection (HCAI) was defined as follows: (1) infection that was not present or incubated at the time of hospital admission; (2) infection related to hospital procedures; (3) infection that the newborn acquires as a result of passing through the birth canal or contact with surgical instrument; and (4) infection occurring in the 30 days following the intervention. Infections that did not meet these criteria were considered to be community acquired [3].

Infections appearing during the first 72 h of life were considered as early type, while those from the 72 h to 30 days of life as late type [3].

The isolates of Gram-negative bacilli that were recovered from different clinical samples in the microbiology laboratories of the hospitals that participated in the research were transferred in Heart Brain Infusion Agar plates, which were then transported to the NRL-HCAI/IPK complying with the biosafety measures, with basic triple packing system recommended by WHO.

In the NRL-HCAI/IPK, species identification was based on API 20 E system (bioMérieux, France) for the identification of enterobacteria and API 20 NE system (bioMérieux, France) for the identification of non-enterobacterial and non-annoying Gram-negative bacilli following the manufacturer's recommendations.

The antimicrobial susceptibility test was carried out with automated system Vitek 2 compact version 6.019 (bioMérieux, Marcy-l'Étoile, France), following the manufacturer's recommendations. The antimicrobials evaluated were as follows: ampicillin, ampicillin/sulbactam, piperacillin/tazobactan, imipenem, meropenem, ceftazidime, cefotaxime, cefepime, gentamicin, amikacin, ciprofloxacin, levofloxacin, phosphomycin and colistin. In the case of colistin, as the Vitek 2 compact system is not recommended to assess susceptibility [8], it was checked with gradient strips (Liofilchem® MIC Test Strip) (Liofilchem S.r.l., Rosetodegli Abruzzi, Italy) based on the Kirby–Bauer method and the interpretation was performed according to the cut-off points defined by 2017 EUCAST standards [9].

For the quality control, the following standard strains were used: *Escherichia coli* ATCC 25922, *Klebsiella pneumoniae* ATCC 35657 and *Pseudomonas aeruginosa* ATCC 27853.

In the analysis and statistical processing of the results, Microsoft Excel 2016 was used for descriptive statistics.

## 3. Results

During the studied period, 61 isolates of Gram-negative bacilli causing neonatal infections were isolated in the sixHavana hospitals among 13,049 live births (0.47%) [10]. Figure 1 shows the clinical forms and origin of infections caused by Gram-negative bacteria. In the cases from the community, the predominant clinical form was urinary tract infection (UTI) (87%). In the healthcare-related infections, sepsis took first place (63.3%), followed by respiratory infection (13.3%). Central nervous system (CNS) infection was not detected.

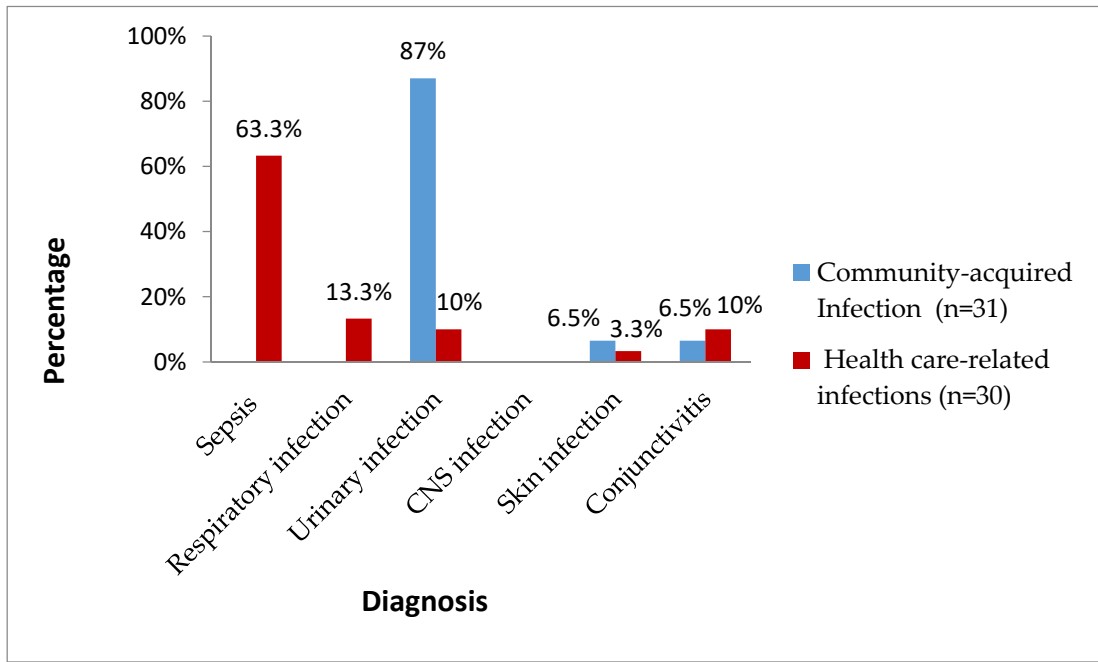

**Figure 1.** Clinical forms and origin of neonatal infections by Gram-negative bacilli. Havana, 2017–2018.

Figure 2 shows types of the infections with Gram-negative bacilli by early or lateonset. Late-onset infections were dominant and included UTI (100%), sepsis (79%), skin infection (100%), conjunctivitis (60%) and respiratory infection (50%).

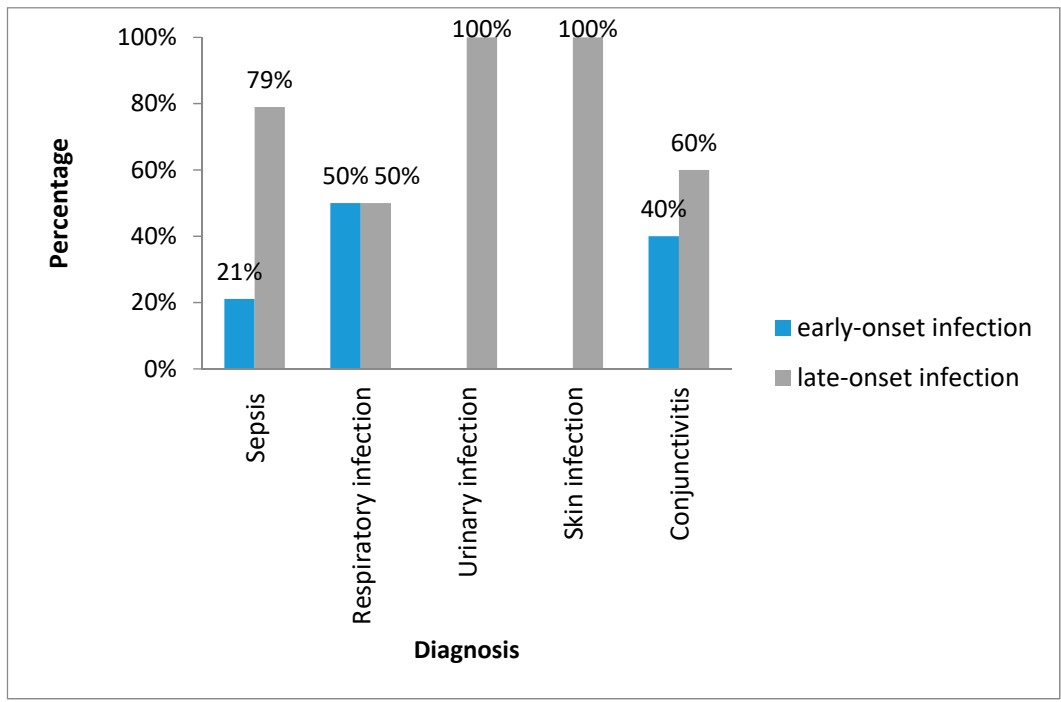

**Figure 2.** Classification of the neonatal infections with Gram-negative bacilli. Havana, 2017–2018.

Table 1 shows the frequency of risk factors in the patients. The use of catheter showed the highest rate (47%) followed by prolonged stay in the neonatology ward (38%), low birth weight (38%) and prematurity (33%). The use of nasogastric tube (25%), parenteral feeding (25%) and invasive mechanical ventilation (23%) were risk factors that also occurred with high frequency.

**Table 1.** Risk factors in neonatal infections with Gram-negative bacilli, Havana, 2017–2018.

| Risk Factors | Total (*n* = 61) | % |
|:---:|:---:|:---:|
| Catheter Use | 29 | 47 |
| Nasogastric tube | 15 | 25 |
| Invasive mechanical ventilation | 14 | 23 |
| Parenteral feeding | 15 | 25 |
| Prior antibiotic therapy | 9 | 15 |
| Prolonged stay in neonatology service | 23 | 38 |
| Genito-Urinary Infection in the last trimester | 10 | 16 |
| Low birth weight | 23 | 38 |
| Prematurity | 20 | 33 |
| Others | 13 | 21 |

Figure 3 shows the distribution of each of the bacterial species identified. The most common Gram-negative microorganism were *E. coli* (47%), followed by *Klebsiella* spp. (26%) and *Enterobacter cloacae* (14%). Only one non-fermenting bacillus that corresponded to *Stenotrophomonas maltophilia* (2%) was isolated, representing an alert because of the therapeutic implications for the management of this germ.

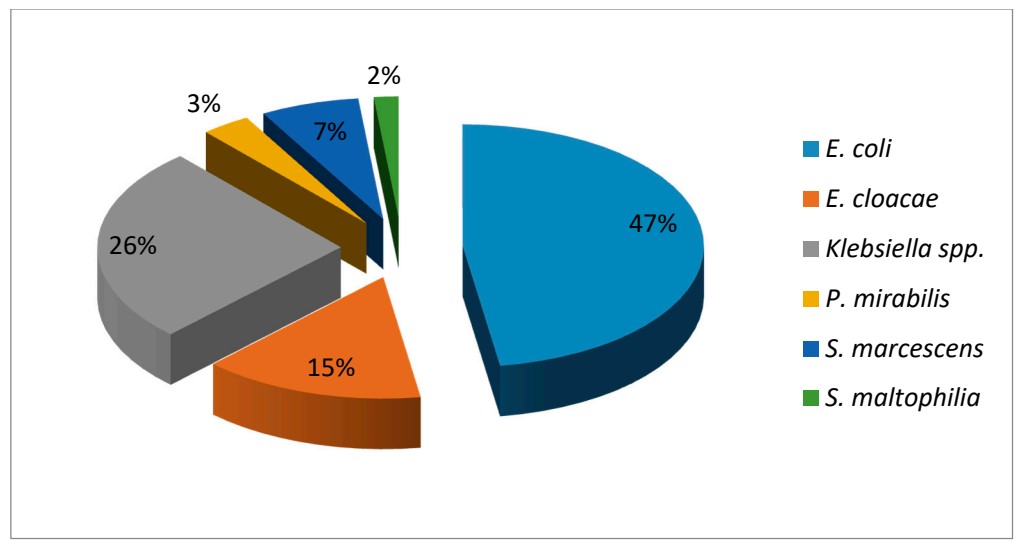

**Figure 3.** Distribution of species of Gram-negative bacilli causing neonatal infections Havana, 2017–2018.

Figure 4 shows resistance rates to β-lactam antimicrobials of different bacterial species. Except for *P. mirabilis*, four Gram-negative species showed high resistance rates to penicillin (>59%) and ceftriaxone (31–75%).

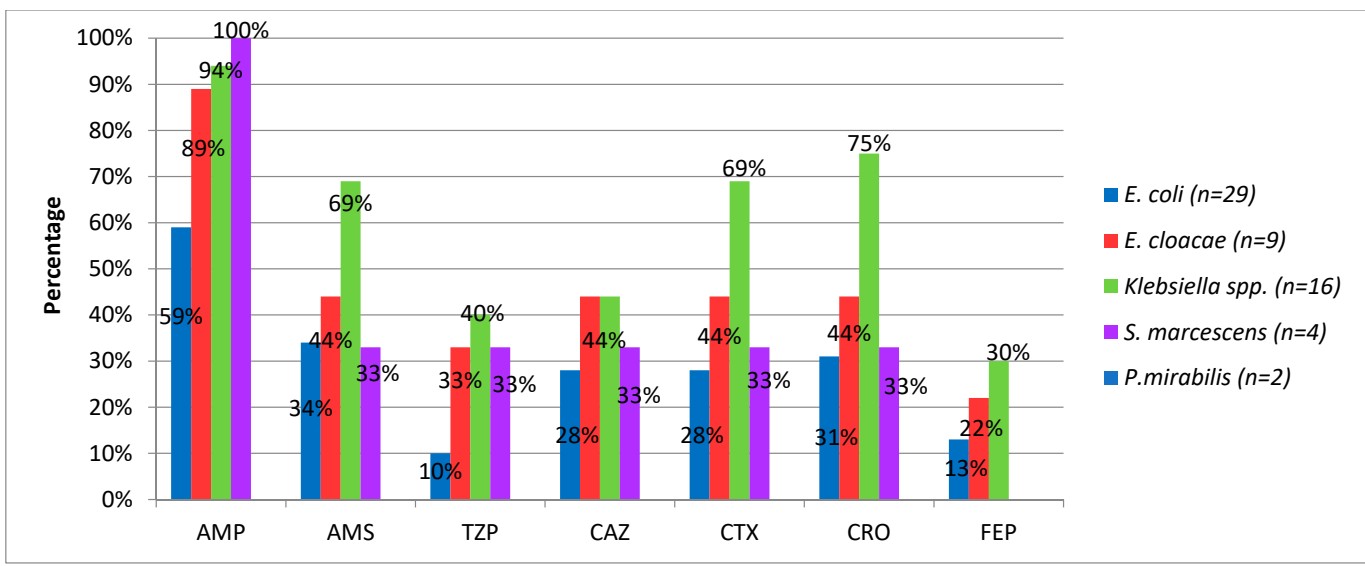

**Figure 4.** Percentage of beta-lactam resistance in Gram-negative bacilli causing neonatal infection. Havana, 2017–2018. AMP, ampicillin; AMS, ampicillin/sulbactam; TZP, piperacillin/tazobactam; CAZ, ceftazidime; CTX, cefotaxime; CRO, ceftriaxone; FEP, cefepime.

Figure 5 shows resistance rates to non-beta-lactam antibiotics. It was found that the resistance to quinolones was above 20% in most identified species, although theseantimicrobials are used in neonates only when alternatives fail in treatment and antibiogram proves sensitivity of this pharmacological group, when assessing the risk benefit relationship.

The resistance rate to the aminoglycosides, which are used as alternatives in synergistic combination with β-lactams, was above 20% in most of the identified species. Amikacin had better in vitro activityperformance. No resistance to colistin and fosfomycin was identified.

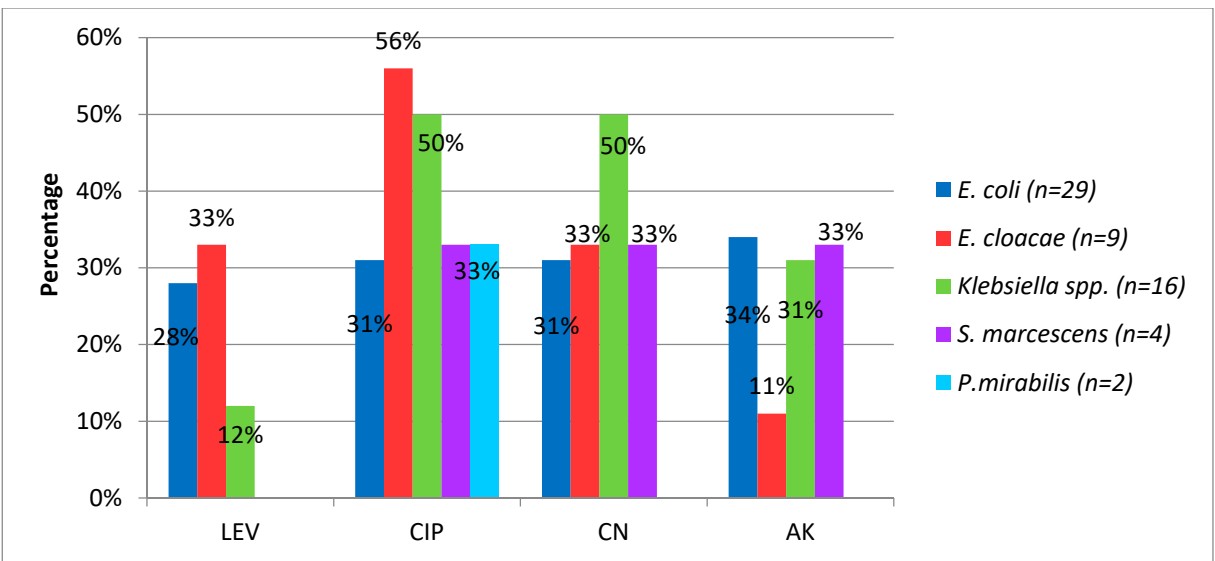

**Figure 5.** Percentage of resistance to non-beta-lactam antibiotics in Gram-negatives bacilli causing neonatal infection. LEV, levofloxacin CIP, ciprofloxacin GM, gentamicin AK, amikacin.

## 4. Discussion

It is estimated that up to 2% of fetuses get infected in uterus and up to 10% of newborns get infected during delivery or during the first month of life [11]. Vertical transmission predominates in newborn infections, as well as infections associated with healthcare, which appear in the first days of birth or during the in hospital stay [11].

Between 33% and 66% of newborns admitted to the Neonatal Intensive Care Unit (NICU) are diagnosed with infection at some time during their stay. Sepsis is one of the most serious manifestations and can also occur as a complication of a localized infection or of another disease of the newborn. It is generally associated with multiple risk factors [11].

In the current research, it is highlighted that UTI represents the highest number of infected cases of community origin. This accords with the findings of Alfonso and collaborators in Chile in 2018, when studying 508 febrile neonates from the community, identifying the UTI (75%) as the main reason for admission [12]. This infection could be related to possible abnormalities of the urinary tract, mainly vesicoureteral reflux [13]. In the present study, this clinical entity was confirmed inonlythreeneonates of the male sex. Hygienic factors also have an influence, since newborns are incontinent, and, due to the proximity of the anus tothe urethra, colonization and ascent of microorganisms to the upper urinary tract can occur [14].

Bloodstream infections followed by respiratory infections are the most frequent infections in the NICU worldwide. The results of the present investigation as well as other studies in Cuba notify this fact. For example, Campo and collaborators in 2011 in Guanabacoa, Cuba, reported predominance of sepsis (54.7%) and lung infection (27.6%) among the 47 live born babies whowere studied [15]. Similarly, Márquez and collaborators in Pinar de Río, Cuba in 2015, in a study including neonates admitted to the NICU, reported sepsis (52%) and respiratory infection (17.7%) [16]. On the other hand, García and collaborators had similar results in Mexico in 2015, with a predominance of sepsis (62.8%) and respiratory infection (19.4%) [17]. These infections are associated with increased mortality rates, immediate and long-term morbidity, prolonged hospital stay and increased cost of care. This is further complicated by the occurrence ofoutbreaks with multi-drug resistant organisms that may continue to occur in NICUs worldwide [18].

It is not always possible to identify the causative agent of neonatal infection because this infection is related to some factors such as the beginning of the first doses of antibiotic management before taking the sample and the specific conditions and aspects when taking the samples themselves [19].

The presumptive diagnosis of neonatal infection is more frequent in early-onset events, while microbiological confirmation is more common in late-onset infections [20]. This could explain the dominance in the current study of late-onset infections, because they are all events confirmed by microbiological tests.

These results arein accordance with previous studies carried out by Pérez and collaborators in Cuba in 2015, who identified a higher percentage of late-onset infections (70%) [21]. Genes and collaborators in Paraguay in 2013, when analyzing infected newborns for 11 years, identified 3.8% of early infection and 96.2% of late infection [22].

In contrast, Alfonso and collaborators reported different results in Cuba in 2016, identifying a predominance of early-onset infections (72.5%) [23]. Similarly, Pérez and collaborators, in a four-year study in Colombia in 2018, identified patients with confirmed neonatal sepsis, of which 70% corresponded to early sepsis [24]. These differences could be related to the fact that in both studies respiratory infections predominated and risk factors were present because of early infection such as maternal chorioamnionitis, intrapartum maternal fever and premature rupture of membranes for more than 18 h. This was not found in the present study.

Since the early 1980s, epidemiological studies have observed a general reduction in early-onset sepsis, probably due to advances in obstetric care and the use of prophylactic intrapartum antibiotics to prevent infections caused by Group B Streptococcus, different fromthe incidence of neonatal late-onset sepsis, which increases with the improved survival of premature infants, indicating the role of hospitalization and life-sustaining medical devices in the pathogenesis of neonatal late-onset sepsis [25].

Multiple factors favor the development of infections in the neonates. However, it is possible to find that the relevance of each of these factors is different [26] in each hospital of different countries and even within the same country.

Vascular access is one of the essential tools for the management of hospitalized newborns, because up to 90% of patients require the administration of parenteral treatment; however, its placement and permanence are also risk factors for neonatal complications [26].

The incidence of neonatal infection is inversely related to the gestational age at birth, which ranges from 46 to 54% in infants under 25 weeks, decreasing to 29% between 25 and 28 weeks and 10% between 29 and 32 weeks of pregnancy [27].

A study published in 2016 undertaken at the Eusebio Hernández Pérez Hospital in Havana, including newborns who developed infection associated with medical care, highlighted that the most relevant risk factors were catheter use (86%), low weight (85%) and prematurity (83%) [28].

A comprehensive review on infections in newborns was published in 2017. The 36 target articles indicate that most of these diseases are related to the unhealthy environment and the hands of healthcare workers anddiscussesthe relevance of each of the factors. Risk is different, depending on the patients, their management and associated morbidities. This review states tht infections are more frequent in infants with prematurity and low weight andwiththe use of catheter and other invasive procedures [29].

Regarding the etiologic agents, our results show the impact of enterobacteria as a cause of infection in neonates, in contrast with the non-isolation of *P. aeruginosa* and *A. baumannii*, which are also Gram-negative bacilli that circulate in neonatology services.In the cases of neonatal infection, it is not always possible to identify the causative agent. This may be related to the fact that the first dose of antibiotic is administered before taking the sample and the specific conditions and aspects present when taking it [30].

Gram-negative bacilli are involved in a wide variety of infections, including 35% of sepsis and 70% of UTIs [20]. The main reservoir of these germs is the gastrointestinal tract where they develop a concentration of up to $10^8$ bacteria per gram of fecal matter [20]. As newborns are incontinent, they permanently pollute their skin and environment. A small contamination with fecal matter can pass thousands of bacteria to the hands of the staff attending the newborns, which favors their transmission to other patients [30].

Specific studies in the province of Santiago de Cuba in 2010 [31] and in Pinar del Río in 2015 [16] reaffirm the *Enterobacteriaceae* family as the Gram-negative bacteria most frequently occurring in neonatology services.A systematic review of neonatal infection in low-income countries was published in 2015. It is evident that enterobacteria are the main Gram-negative bacilli involved in these pathologies, with a superiority of *K. pneumoniae* (12–33%), *E. coli* (9–19%) and *Enterobacter* spp. (8 to 10%) [32].

The results of the present research differ from those obtained by Crivaro and collaborators in Italy in 2015 [33] when studying neonates treated in NICU for a year. They reported superiority of *Pseudomonas aeruginosa* and *Acinetobacter* spp. and thirdly *Klebsiella pneumoniae*. These differences could be related to the variations in the etiology of the bacterial cause of neonatal infection according to country, hospital, time of year and patient risk factors [33].

In general, in our study, the percentage of resistance to β-lactams, which are first-line drugs in the treatment of neonatal infection due to their low toxicity and broad spectrum, was above 20% in most of the identified species. We think this is due to the production of Extended Spectrum Betalactamases (ESBL), which is the main mechanism of resistance in enterobacteria to this pharmacological group and that has previously been notified in Cuba [34].

The study highlights that *P. mirabilis* was the only species that showed no resistance to the β-lactam group.

Antimicrobial resistance is a worldwide problem and a constant concern of the international scientific community [35]. Numerous prevention strategies have been proposed, including strict control of antimicrobial use, appropriate combinations and rotation of antibiotics.However, in recent years, resistant microorganisms havebeen increasing, which leads to the search for new antibiotics and the recovery of other drugs used long time ago [35].

β-lactams, being the group of ATMs most used worldwide, provide the highest resistance rates. In NICUs, the selective pressure exerted on the microbial populations with the use of these drugs causes an increase in colonization by bacteria gradually becoming resistant to the usual therapy [36].

Specific studies in Cuba by Santisteban and collaborators in 2014 [37] and a more recent study by Rodríguez and collaborators in Havana in 2016 [28] confirm high percentages of resistance to third-generation cephalosporins in enterobacteria isolated in neonates. In Colombia (Vélez et al., in 2014), when analyzing the susceptibility profile in Gram-negative bacteria isolated in a pediatric population, a study indicated high percentages of resistance tothird-generation aminopenicillins and cephalosporins in *K. pneumoniae*, *E. cloacae* and other enterobacteria [38]. González in Colombia in 2014, when analyzing 43 epidemiological surveillance reports with information on the resistance of isolated enterobacteria in hospitals with a neonatology service, pointed out that *K. pneumoniae* and *E. cloacae* were the species with the highest resistance to β-lactams with values of 35–45% [39].

A Cuban study published in 2016 reported high levels of resistance in enterobacteria to quinolones and aminoglycosides [28].

Foreign authors in Guadalajara in 2015 [40] and Nepal in 2018 [41], when analyzing the bacteriological profile and susceptibility patterns of common isolates in patients with neonatal sepsis, described high percentages of resistance in the *Enterobacteriaceae* family to aminoglycosides, although they pointed out amikacin is the least affected.

In the current research, only one *S. maltophilia* isolate was presented, in which the susceptibility to β-lactams, aminoglycosides and colistin was not evaluated, because this germ presents intrinsic resistance to these pharmacological groups. The recommended alternatives of choice against infections by this germ are sulfa drugs and tetracyclines, which are contraindicated in neonates [42]. One of the options to be used in the managing of this germ is levofloxacin, which is recommended in neonates as this germ fortunately proved to be sensitive to it in the current study.

*S. maltophilia* is widely distributed in the environment; it is an in-hospital pathogen and the frequency of infections iscurrentlyincreasingin intensive care units due to this germ, which is related to bacteremia, endocarditis andrespiratory tract infections dueto mechanical ventilation, tracheotomy or prolonged antibiotic treatment. It is also associated with infections of the central nervous system in neonates and young children where it appears spontaneously [42].

Abbasi and collaborators and Sirvan and collaborators, both in Turkey, in 2009 and 2015, respectively, detected *S. maltophilia* in several neonates with respiratory infection and sepsis; the antibiotic administration to these infections was implemented with quinolones and the lethality was over 63% [43,44]. This proves that the antibiotic treatment of *S. maltophilia* infections is still a challenge.

## 5. Conclusions

This research is a response to neonatal infections, which areone of the priorities in the Maternal and Child Program in Cuba. It demonstrated the importance of Gram-negative bacilli as the etiological agent in this population, which imposes the enhancement of prevention and control measures. Early and specific diagnosis of infections with adequate antimicrobial treatment is essential, taking into account the limited treatment options in the neonate in the face of increased antimicrobial resistance and the high mortality rate.

**Author Contributions:** Conceptualization, D.Q., R.Á. and E.d.L.C.L.; methodology, D.Q., M.S.A., E.d.L.C.L. and R.Á.; formal analysis, A.O.,Y.C., M.S.A. and N.K.; investigation, A.O. and Y.C.; resources, D.Q.; data curation, A.O.; writing—original draft preparation, A.O. and D.Q.; writing—review and editing, D.Q., M.S.A. and N.K.; visualization, A.O., E.d.L.C.L. and D.Q.; supervision, D.Q.; project administration, D.Q.; and funding acquisition, D.Q. All authors have read and agreed to the published version of the manuscript.

**Funding:** This work was conducted with support from Public Health Ministry of Cuba.

**Institutional Review Board Statement:** The study was conducted according to the guidelines of the Declaration of Helsinki, and approved by the Institutional Ethics Committee of The Tropical Medicine Institute "Pedro Kourí" (CEI-IPK 39-17; date of approval: April 2017).

**Informed Consent Statement:** "Not applicable" for studies not involving humans.

**Data Availability Statement:** The data presented in this study are openly available on the website www.ipk.sld.cu (accessed on 15 February 2021).

**Acknowledgments:** We are grateful to all neonatologists and microbiologists whocontributedto thesurveillanceof neonatal infection by Gram-negative bacilli in Gyneco-Obstetric and Pediatric Hospitals of Havana.

**Conflicts of Interest:** The authors declare no conflict of interest.

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
