# Peer review of "Characterization of Neonatal Infections by Gram-Negative Bacilli and Associated Risk Factors, Havana, Cuba"

_2036-7449, doi:10.3390/idr13010025_

Round 1
Reviewer 1 Report
The article is devoted to an important task - saving the life of newborns and analyzing the causes of infection and death due to the ineffectiveness of antibiotic therapy. The authors examined infections in newborns in six hospitals in Cuba. The causative agents of infections, the effectiveness and the very possibility of treatment with antibiotics, as well as the zones of organisms' damage have been analyzed. Resistance to several of the currently most applicable antibiotics is reviewed. The authors believe that further monitoring on a systemic basis of infectious agents in newborns and the effectiveness of antibiotic therapy is an important task. The article analyzes related literature on this topic, mainly from the countries of the region. An overall assessment of the routes of infection and the effectiveness of antibiotic therapy is given.
The disadvantage of this manuscript is some carelessness of the text. Authors on different pages write differently the designation of a group of bacteria, a common terminology is needed, it is desirable to write Gram-negative, which is most recognized.
The specific names of bacteria should be italicized, which is not everywhere in the text. For example, on lines 265. On the contrary, do not italicize ”sp.” And “spp.”.
The plural bacillus is written as “bacilli”, not “bacillis”. Sometimes they write “bacillis” in the text.
The article can be published after careful correction of the text.
Author Response
"Please see the attachment."

Reviewer 2 Report
The manuscript describes the incidence, species, infection type and antimicrobial resistance profiles of clinical samples of neonates in Cuba from 2017-2018. While the findings have importance, the presentation of the work needs improvement prior to publication.
- There were a variety of writing and grammatical problems including: (1) Sentence structure problems, starting in the abstract including subject verb agreement like in “It is known an increase in neonatal infection”, “The aim of study 14 was provide evidences on” and “It was carried out a descriptive cross-sectional 16 investigation”; (2) multiple spacing issues in the paper, and (3) things like the phrase Gram-negative refers to a name and the “gram” must be capitalized (which was written with a hyphen in the abstract and without one in various places in the text). coli must be italicized. The authors need to employ se of an English Language Editing service to help with the writing of their article.
- The methods selected and the written section needs improvement, in that more than one citation for the methods should be available, and if the methods don’t follow standard procedures, the authors need to justify why they made that choice. While the number of specimen studied were described, it would be valuable to indicate how many births were considered over this same time in these hospitals to give a sense of the likelihood of infections.
- In the beginning of the results section the authors also suggest that many patients would be administered antimicrobials, which would limit the possible number of samples, but without this data being provided and analyzed, this statement is just speculation (unless there are references to support this idea).
- Figure 3 states it breaks apart data by gender and species, but there is no gender breakdown shown.
- Also, Excel isn’t a good statistics analysis program. And the plot style for Figures 4 & 5 aren’t allowing for clear presentation of the data.
- Line 185 and beyond, he authors I believe refer to UTI when they say ITU.
- The authors in lines 185-268 describe many difference when compared to other findings from different researchers, but this is done in a way that doesn’t delve into the data or really feel much more than a list of references. This discussion needs considerable improvement. The section of antimicrobial resistance read better than the incidence of different bacteria and the infection types.
- The Conclusions section is overly succinct. Major finding should be described.
Author Response
"Please see the attachment."
